# Transcriptional and Post-Transcriptional Polar Effects in Bacterial Gene Deletion Libraries

André Mateus,[a] Malay Shah,[a] Johannes Hevler,[a] Nils Kurzawa,[a,b] Jacob Bobonis,[a,b] Athanasios Typas,[a] Mikhail M. Savitski[a]

[a]European Molecular Biology Laboratory, Genome Biology Unit, Heidelberg, Germany
[b]Collaboration for Joint PhD Degree from EMBL and Heidelberg University, Faculty of Biosciences, Heidelberg, Germany

**ABSTRACT** Single-gene deletions can affect the expression levels of other genes in the same operon in bacterial genomes. Here, we used proteomics for 133 *Escherichia coli* gene deletion mutants and transcriptome sequencing (RNA-seq) data from 71 mutants to probe the extent of transcriptional and post-transcriptional effects of gene deletions in operons. Transcriptional effects were common on genes located downstream of the deletion and were consistent across all operon members, with nearly 40% of operons showing greater than 2-fold up- or downregulation. Surprisingly, we observed an additional post-transcriptional effect that leads to the downregulation of the gene located directly downstream of the targeted gene. This effect was correlated with their intergenic distance, despite the ribosome binding site of the gene downstream remaining intact during library construction. Overall, the data presented can guide future library construction and highlight the importance of follow-up experiments for assessing direct effects of single-gene deletions in operons.

**IMPORTANCE** Single-gene deletion libraries have allowed genome-wide characterization of gene function and interactions. While each mutant intends to disrupt the function of a single gene, it can unintentionally target other genes, such as those located in the same operon as the deletion. The extent to which such polar effects occur in deletion libraries has not been assessed. In this work, we use proteomics and transcriptomics data to show that transcript level changes lead to nearly 40% of deletions in operons affecting the protein levels of genes located downstream by at least 2-fold. Furthermore, we observed a post-transcriptional effect on the gene located directly downstream of the deletion. These results can guide the design of future gene deletion libraries and emphasizes the importance of follow-up work when linking genotypes to phenotypes.

**KEYWORDS** Keio collection, knockout, polar effects, proteomics, single gene deletion

Genome-scale gene deletion (1–5) or knockdown (4, 6, 7) libraries have revolutionized high-throughput approaches for characterizing gene function and interactions in microbes (8–11). These libraries intend to disrupt the function of single genes but can inadvertently affect other genes. For example, short nucleotide sequences used to target a specific gene in CRISPR-based methods can display high complementarity to other regions of the genome (off-target effects) (12). Likewise, the expression of genes proximate to the targeted gene can be affected in CRISPR-based systems (6, 13, 14), transposon mutagenesis (15, 16), or when making targeted deletions via homologous recombination (1, 2, 17). Such polar effects are particularly prevalent in bacteria because of the operon organization, the presence of overlapping coding sequences, *cis*-acting ribosomal stalling (18, 19), sporadic translation past stop codons (20), and cotranslational protein complex assembly (21, 22). This can lead to erroneous attributions of gene functions, since observed

Address correspondence to Athanasios Typas, athanasios.typas@embl.de, or Mikhail M. Savitski, mikhail.savitski@embl.de.

Widespread polar effects in bacterial gene deletion libraries.

phenotypes attributed to the targeted gene can be caused by expression changes of downstream genes.

The *Escherichia coli* single gene deletion library, known as the Keio collection, was carefully designed to avoid both off-target and polar effects (1, 2, 17). For the former, all gene deletion mutant clones were confirmed by PCR for the correct genomic location of the antibiotic cassette. For the latter, the antibiotic resistance cassette was designed to include a weak promoter in front of the kanamycin resistance gene with no transcriptional terminator and was placed in the same orientation as the gene deleted to minimize any transcriptional polar effects. In addition, the last 21 nucleotides of the deleted gene were kept to preserve the ribosome binding site (RBS), generally located at −7 to −12 nucleotides of the start codon (23), and thus avoid protein translation polar effects. Finally, flippase recognition target (FRT) sites were introduced to allow the excision of the antibiotic resistance cassette in order to completely mitigate any downstream transcript level changes. This excision also produces a small in-frame peptide (34 amino acids) (1) that should ensure translational coupling, in which translating ribosomes can continue translating downstream genes (24).

We recently profiled proteome-wide protein levels and thermal stability of 121 mutants (11) of the Keio collection (1, 2, 8) which still retained the antibiotic resistance cassette. In this work, we explore these and other available transcriptional data for Keio mutants (25) and generate new data to uncover both transcriptional and post-transcriptional effects on genes located downstream of the deleted gene. Transcriptional effects led to a >2-fold consistent up- or downregulation of nearly 40% of all downstream genes. We also observed an unexpected post-transcriptional effect on the gene located directly downstream of the deleted gene. By making a series of new constructs and profiling their impact on global protein expression, we attributed this effect to translation initiation problems due to the distance of the cassette and the start of the downstream gene. In practical terms, this work highlights the importance of carefully interpreting results from genetic approaches and genome-wide mutant libraries, showcases the importance of measuring protein levels in order to facilitate gene function associations, and provides guidance for future single-gene deletion design.

## RESULTS AND DISCUSSION

**Keio deletion mutants exhibit prevalent transcriptional polar effects.** Using multiplexed quantitative proteomics (26), we previously profiled the proteomes of 117 single-gene deletion mutants from the Keio collection in duplicates (11). Here, we measured the proteomes of an additional 16 mutants to increase the proportion of mutants on operons (see Table S1 in the supplemental material). Of these 133 gene deletions, 84 were in operons containing at least another gene for which we could measure protein levels (Fig. 1a). We observed that the protein levels of genes upstream of the deletion were largely unaffected compared to those in the wild type, while the expression levels of genes downstream were commonly affected (Fig. 1b). Downstream genes behaved largely consistently within the operon (see Fig. S1). Many operons were upregulated (18% [6/34]) or downregulated by at least 2-fold (21% [7/34])—here we ignored the gene directly downstream of the deletion (see below for specific effects on this gene) and calculated the median of all other downstream genes within the operon. The same picture held true at the mRNA level, with 4/11 of operons with median transcript levels increased or decreased by 2-fold (Fig. 1b; Fig. S1) when reanalyzing data from a recent study on the transcriptome of 71 Keio mutants (25)—which included 27 deletions in operons with at least another gene for which it was possible to measure transcript levels. These effects at the protein and transcript levels were evident even when excluding deletions of genes that may regulate the expression of their own operons (see Fig. S2). These polar effects were absent from deletions of genes that are not part of the same operon (see Fig. S3). Hence, introducing a kanamycin resistance cassette can lead both to an increase (up to 4-fold) and to a decrease (up to 8-fold) in expression of

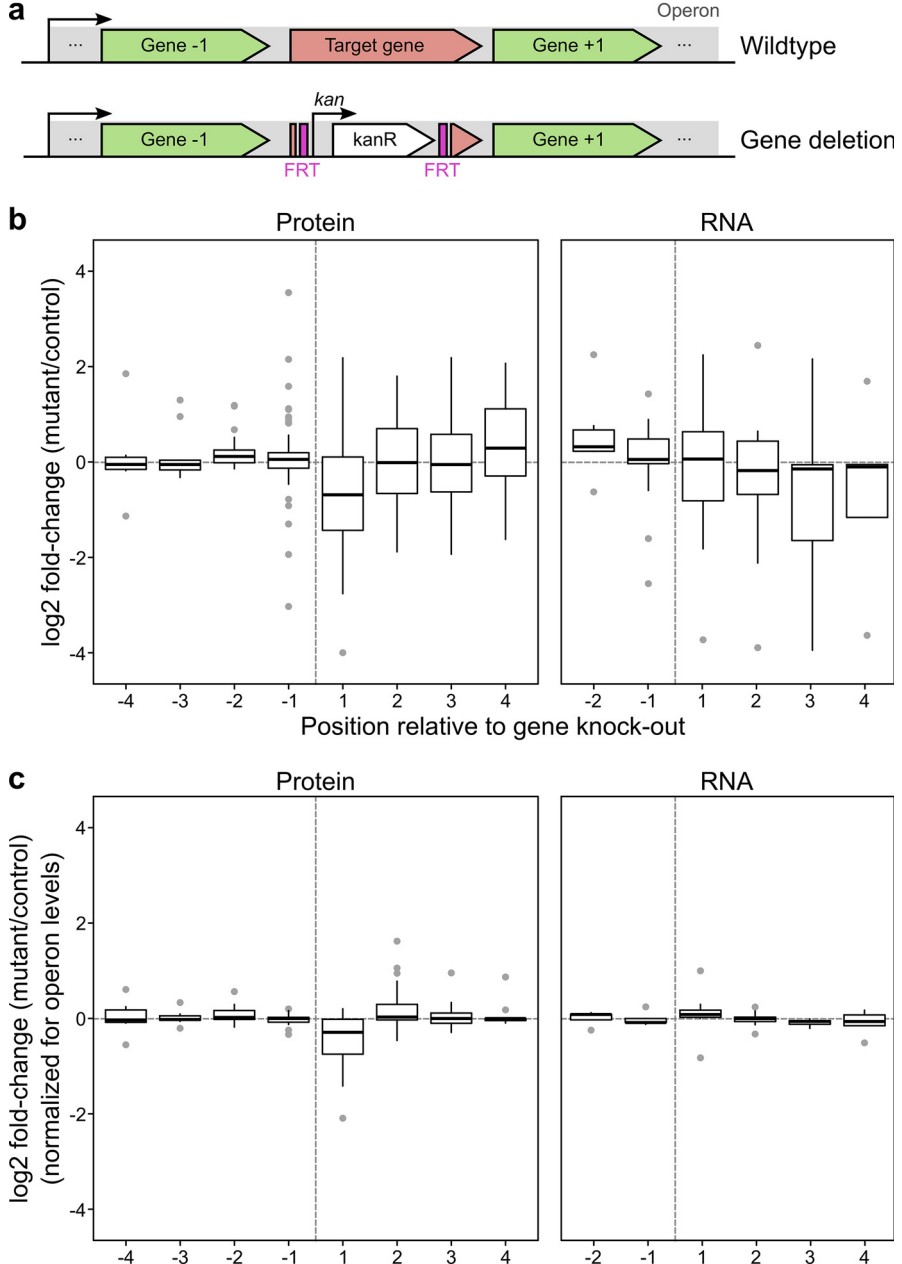

**FIG 1** Transcriptional and post-transcriptional polar effects of gene deletions in *E. coli* operons. (a) Schematic representation of deletion of a target gene that is present in an operon. In the Keio collection (1), the target gene is replaced by a kanamycin resistance cassette flanked by Flp-recognition sites (FRT), preserving the start codon (first 3 nucleotides) and the last seven codons (last 21 nucleotides). (b) Deletion of a gene in an operon leads to transcriptional differences (up- or downregulation) of genes located downstream of the deleted gene. Data represent $\log_2$ fold change of each gene relative to control (see Materials and Methods for details) as a function of their position in the operon relative to the deleted gene. Center lines in box plots represent the medians, box boundaries indicate the upper and lower interquartile range (IQR), and whiskers correspond to most extreme values or to 1.5-fold of IQR if the extreme values are above this cutoff. Protein (84 deletion mutants): $n_{-4} = 6$, $n_{-3} = 9$, $n_{-2} = 20$, $n_{-1} = 49$, $n_1 = 61$, $n_2 = 33$, $n_3 = 20$, $n_4 = 10$. RNA (27 deletion mutants) (25): $n_{-2} = 7$, $n_{-1} = 19$, $n_1 = 18$, $n_2 = 10$, $n_3 = 6$, $n_4 = 5$. (c) As in panel b but with protein or transcript abundance corrected for the median upstream or downstream abundance changes of protein or transcript in the same operon, including only mutants in operons for which at least two genes can be quantified upstream or downstream. Protein (47 deletion mutants): $n_{-4} = 6$, $n_{-3} = 9$, $n_{-2} = 18$, $n_{-1} = 18$, $n_1 = 31$, $n_2 = 32$, $n_3 = 19$, $n_4 = 10$. RNA (14 deletion mutants) (25): $n_{-2} = 6$, $n_{-1} = 6$, $n_1 = 11$, $n_2 = 10$, $n_3 = 6$, $n_4 = 5$.

downstream genes in an operon. We reasoned that these changes were primarily transcriptional, as they occurred both at the mRNA and protein levels, and mRNA and protein level changes of operon genes were concordant for the six mutants for which we had both datatypes ($r = 0.63$, $P = 0.011$, $n = 15$) (see Fig. S4).

The kanamycin resistance promoter caused the overexpression of downstream genes, as the effect was abolished when excising the cassette from 16 mutants representative of the initial set of 84 mutants probed (compare Fig. S5a with Fig. 1b) and reprofiling their proteome (Fig. S5b; Table S1). In addition, upregulated operons tended to have lower protein abundance when unperturbed ($r = -0.31$, $P = 0.071$, $n = 34$) (see Fig. S6), supporting the idea that the kanamycin promoter is stronger than the natural promoter in these cases. In contrast, excising the cassette did not impact the downregulation (Fig. S5b); so, in these cases, expression changes are likely due to the deletion of intragenic promoters or transcriptional elements or a less stable new mRNA.

In summary, the replacement of a gene by the kanamycin resistance cassette leads to widespread transcriptional up- or downregulation of genes located downstream and in the same operon as the deleted gene.

**Polar post-transcriptional effects on the gene directly downstream of the deletion.** Interestingly, we noticed that the gene directly downstream of the deletion exhibited an overall additional downregulation at the protein level (median = 1.62-fold lower than wild type, $P = 6.9 \times 10^{-5}$ Wilcoxon rank sum test) but not at the mRNA level ($P = 0.93$ Wilcoxon rank sum test) (Fig. 1b). This decrease in the protein level of the downstream gene was the only significant change ($P = 9.1 \times 10^{-5}$ Wilcoxon rank sum test) remaining after normalizing out the general effect on expression for upstream and downstream genes (Fig. 1c). This effect was still present in the 16 mutants for which the kanamycin cassette was excised (Fig. S5b).

Thus, this strongly suggests that the overall downregulation of the gene located directly downstream of a deletion is post-transcriptional. Since the Keio mutants have been designed to avoid obvious sources of translational polarity (e.g., deletion of ribosomal binding site), we decided to look further into the source of this effect.

**Problems with translation initiation partially explain post-transcriptional polar effects in Keio mutants.** To explore the causes of downregulation of the gene located directly downstream of the deletion, we first wondered if the FRT sequences, due to the complementary nature of their sequence, could lead to mRNA folding that could hinder translation initiation for the downstream gene. We selected three mutants, two of which showed downregulation of the gene directly downstream ($\Delta fruB$ and $\Delta pabC$) and one that did not ($\Delta yecN$), and built different versions of the deletion in which the FRT sites were removed from either or both sides of the resistance cassette (see Fig. S7a). The levels of the gene located directly downstream of the deletion remained similar to the level for the traditional cassette in all cases (Fig. S7b).

Next, we observed a trend between how much the gene directly downstream of the deleted gene is downregulated (after correcting for transcriptional effects) and their intergenic distance—with genes that were at a closer distance showing stronger downregulation ($r_S = 0.62$, $P = 2.5 \times 10^{-4}$, $n = 30$) (Fig. 2a). Thus, we investigated if this effect was alleviated by expanding the region at the end of the deleted gene (i.e., the 21 nucleotides that ensure that the ribosome binding site is kept intact [1]) to 42 and 99 nucleotides for five deletion mutants ($\Delta surA$, $\Delta oppB$, $\Delta pabC$, $\Delta hisG$, and $\Delta rfbB$) (Fig. 2b). Although the protein levels of the downstream gene of $\Delta pabC$, $\Delta surA$, and $\Delta oppB$ moved closer to wild-type levels when the leftover region was increased to 99 nucleotides, the recovery was not correlated with the increase in the number of nucleotides ($r_S = 0.063$, $P = 0.65$, $n = 55$) (Fig. 2c, left). Excising the kanamycin resistance cassette in these mutants further recovered the levels of the gene downstream of $\Delta hisG$ for the longest construct (Fig. 2c, right) and improved the overall correlation between the size of the leftover C-terminal sequence in the deleted gene and the recovery from downregulation ($r_S = 0.38$, $P = 0.024$, $n = 55$).

Overall, these results suggest that translation initiation problems, a lack of translational coupling, or both, depending on the deletion, partially explain why the levels of the gene

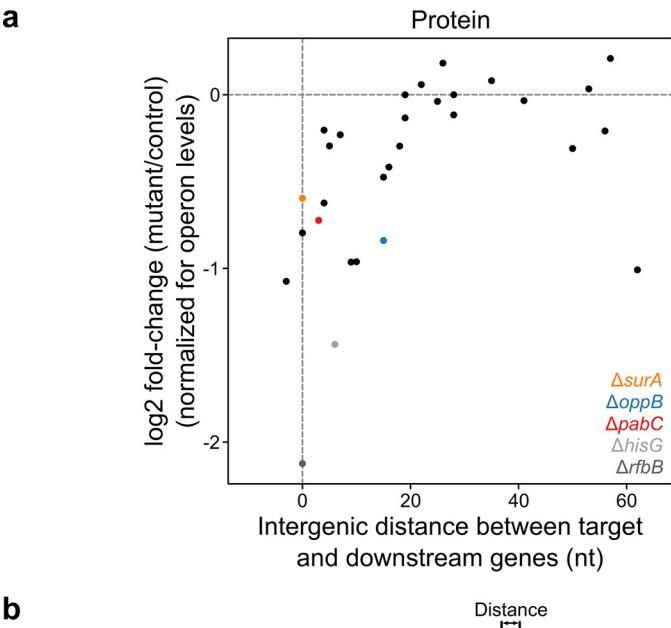

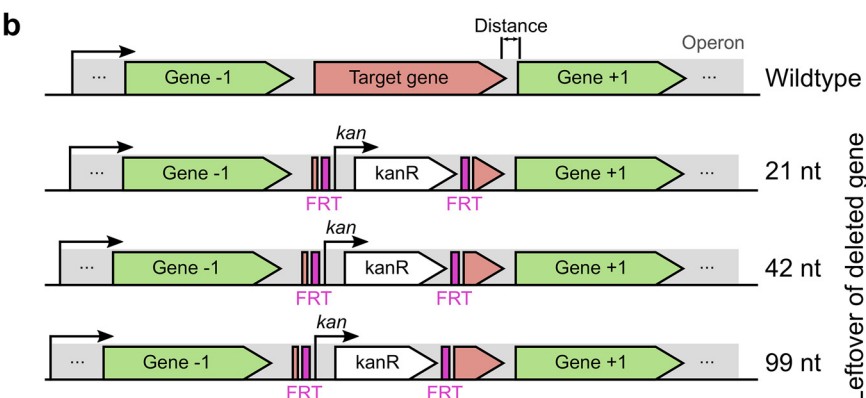

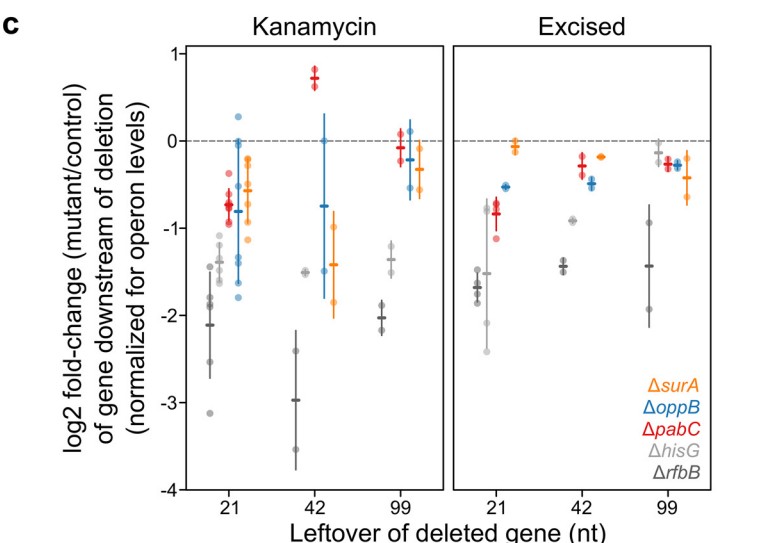

**FIG 2** Effects of extending the leftover C terminus of a deleted gene. (a) Genes that have their start codon closer to the stop codon of the deleted gene are downregulated more at the protein level than genes that are at a greater distance. Note that antibiotic resistance cassette is inserted at the −21 position. Data show means of log$_2$ fold change of protein levels for each gene relative to that for the control corrected for the median downstream abundance changes of proteins in the same operon. (b) Schematic representation of distance between the stop codon of the deleted gene and start codon of the gene downstream (wild type) and mutants built to evaluate the effects of

downstream of a deletion are affected. Furthermore, it seems that translational coupling is affected by the size of the gene being translated, with 34 residues not being enough for most genes (as seen above for a larger number of mutants) (Fig. S5b), but increases to 41 or 60 residues, resulting in alleviation of these effects for most of the deletions studied.

**Conclusions.** In conclusion, single-gene deletion libraries in bacteria, even when carefully designed (such as the Keio collection), show widespread polar effects. Nearly 40% of the gene deletions profiled here showed >2-fold up- or downregulation of transcripts levels of genes located downstream of a deleted gene. Upregulation is likely caused by the presence of the promoter of the resistance cassette, which boosts expression of low-expressed genes. In contrast, the downregulation might be caused by disturbances to intraoperon regulatory sequences or mRNA stability of highly expressed genes. We also showed a post-transcriptional effect on the gene that is located directly downstream of the deleted gene, which generally showed lower protein levels than the rest of genes of the operon. This effect was correlated with the intergenic distance between the deleted gene and the downstream gene. Our follow-up work suggests that problems with translation initiation or lack of translational coupling lead to this effect, but further studies will be required to properly understand the underlying mechanism. Future library design could be improved by increasing the C-terminal scar of the deleted gene. When using existing libraries, it is important to be aware that any such polar effects can lead to the misinterpretation of phenotype-genotype associations, and complementation experiments (in which the knocked-out gene is ectopically expressed to restore the phenotype) are important follow-ups to distinguish direct from indirect effects. As an example, we have recently found that the *ybaB* deletion mutant is sensitive to UV, not because cells lack YbaB but because of the low levels of the adjacently encoded RecR (11). More broadly, this work highlights the added value of characterizing *E. coli* mutants at the proteome level, which is now possible in a fast manner by using multiplexed quantitative mass spectrometry (11, 27, 28).

## MATERIALS AND METHODS

**Strains.** All mutants used in this study have been made in the *E. coli* BW25113 strain background. When available, mutants were used directly from the Keio collection using two independent clones (with the exception of Δ*atpD*, Δ*atpE*, Δ*clpP*, Δ*dedD*, and Δ*rfaC*, for which the same clone was used in duplicates) (1, 17) (see Table S2 in the supplemental material). The remaining strains were built following the strategy of Baba et al. (1) using the primers described in Table S2. Briefly, PCR fragments were generated from the pKD13 plasmid and electroporated into the BW25113 strain, and two colonies were isolated and verified by PCR. Kanamycin resistance cassette excision was performed using the protocol by Datsenko and Wanner (17) using the pCP20 plasmid and further verified by PCR.

**Proteomics analysis.** Protein level measurements that did not originate from Mateus et al. (11) were performed as follows: cells were grown to an optical density at 578 nm (OD$_{578}$) of ~0.5, washed with phosphate-buffered saline (PBS), and lysed by the addition of 2% SDS followed by an incubation at 95°C for 10 min.

Preparation of samples for mass spectrometry was performed as previously described (11, 27). Briefly, samples were digested into peptides using a modified SP3 protocol (29, 30), and peptides were labeled with tandem mass tag (TMT) reagents (Thermo Fisher Scientific) and fractionated using high pH fraction. These samples were then analyzed by liquid chromatography-tandem mass spectrometry (LC-MS/MS) and searched against the *E. coli* (strain K-12) UniProt FASTA (proteome identifier [ID] UP000000625) using isobarQuant (31) and Mascot 2.4 (Matrix Science).

**Calculation of protein level changes.** For every protein in every mutant, we calculated the relative protein concentration to that in the wild type. For this, we started by normalizing the signal sum intensities of each TMT channel with *vsn* (32). Then, for every protein, we calculated the ratio of the signal sum intensity of each mutant to the median signal sum of the same protein in all the mutants in the same mass spectrometry experiment (for data originating from Mateus et al. [11], we used the first two temperatures analyzed for each mutant). Fold change data from replicates were averaged. The code to generate these data is available at https://github.com/andrenmateus/gene_deletion_effects_on_operons.

**FIG 2** Legend (Continued)
increasing the leftover codons of the deleted gene. (c) Increasing the number of codons left over in the deleted gene partially rescues the levels of the gene downstream. Mutants that retain the kanamycin resistance cassette are shown on the left, and mutants for which this was excised are shown on the right. Horizontal bars show means and error bars show standard deviations of log$_2$ fold change of protein levels of each gene relative to control corrected for the median downstream abundance changes of proteins in the same operon.

**Transcriptomics data analysis.** We downloaded the WIG files from the data set GSE129161 deposited in Gene Expression Omnibus. For each mutant, the counts at each position were summed from the forward and reverse tracks (from the "all" files). Then, for each gene, we retrieved the chromosome positions from the annotations in NCBI GenBank accession number NC_000913.2, summed up all counts within those positions for each mutant, and divided this by the gene length. We removed all cases in which this count value normalized by gene length was <1. For each gene in each mutant, we then calculated the relative transcript levels to that in the wild type by calculating the ratio between the count value normalized by gene length by the median count value normalized by gene length across all mutants.

**Normalization of transcriptional effects on the operon.** We retrieved the operon structure from Ecocyc v21.1 (https://ecocyc.org/) (33). Then, for each gene of interest (knockout), we calculated the median $\log_2$ fold change (protein or transcript) for all genes in the same operon located upstream of the gene of interest and subtracted this value from the $\log_2$ fold change for every gene located upstream of the gene of interest. We repeated the same procedure for genes located downstream of the gene of interest.

**Data availability.** The mass spectrometry proteomics data have been deposited to the ProteomeXchange Consortium via the PRIDE partner repository with the data set identifier PXD023945. We further retrieved data from the PRIDE repository with the accession number PXD016589. RNA-seq data were collected from the Gene Expression Omnibus (accession number GSE129161).

**Code availability.** The code to process raw mass spectrometry data (available at PRIDE partner repository with the data set identifiers PXD023945 and PXD016589) and the $\log_2$ fold change of mutant versus wild type (Table S1) is available at https://github.com/andrenmateus/gene_deletion_effects_on_operons.

## SUPPLEMENTAL MATERIAL

Supplemental material is available online only.

**DATA SET S1**, XLSX file, 0.1 MB.
**FIG S1**, TIF file, 0.7 MB.
**FIG S2**, TIF file, 0.5 MB.
**FIG S3**, TIF file, 0.6 MB.
**FIG S4**, TIF file, 0.3 MB.
**FIG S5**, TIF file, 1.2 MB.
**FIG S6**, TIF file, 0.3 MB.
**FIG S7**, TIF file, 0.9 MB.
**TABLE S1**, XLSX file, 18.9 MB.
**TABLE S2**, XLSX file, 0.1 MB.

## ACKNOWLEDGMENTS

This work was supported by the European Molecular Biology Laboratory. A.M. was supported by a fellowship from the EMBL Interdisciplinary Postdoc (EI3POD) program under Marie Skłodowska-Curie Actions COFUND (grant number 664726). A.T. is supported by an ERC consolidator grant, uCARE.

We thank the proteomics core facility at EMBL for expert help.

A.M., M.M.S., and A.T. designed the study. A.M., M.S., J.H., and J.B. performed the experiments. A.M. and N.K. performed the data analysis. A.M., A.T., and M.M.S. drafted the manuscript, which was reviewed and edited by all authors. A.T. and M.M.S. supervised the study.

We declare no competing interests.

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
