## [Reviewer comments · mSystems]

Transcriptional and post-transcriptional polar effects in bacterial gene deletion libraries

André Mateus, Malay Shah, Johannes Hevler, Nils Kurzawa, Jacob Bobonis, Athanasios Typas, and Mikhail Savitski

Corresponding Author(s): Mikhail Savitski, European Molecular Biology Laboratory

Review Timeline:

Submission Date:	June 22, 2021
Editorial Decision:	July 14, 2021
Revision Received:	July 28, 2021
Accepted:	August 18, 2021

Editor: Laura Sanchez

Reviewer(s): The reviewers have opted to remain anonymous.

Transaction Report:

DOI: <https://doi.org/10.1128/mSystems.00813-21>

July 14, 2021

Dr. Mikhail Savitski
European Molecular Biology Laboratory
Heidelberg
Germany

Re: mSystems00813-21 (Transcriptional and post-transcriptional polar effects in bacterial gene deletion libraries)

Dear Dr. Mikhail Savitski:

Thank you for submitting your manuscript to mSystems. We have completed our review and I am pleased to inform you that, in principle, we expect to accept it for publication in mSystems. However, acceptance will not be final until you have adequately addressed the reviewer comments.

I find this paper appropriate for an observation and will be of high interest to the readership given the popularity of these libraries within microbiology. While Reviewer 1 would like mechanistic insights, I believe this is beyond the scope of an observation, however if a model could be proposed that may help alleviate some of the concerns, but I will leave this to the authors to decide if they feel this is appropriate. The suggestions from both reviewers, especially in clarity of the experiments and data presented would strengthen the observation and the authors are strongly encouraged to address these comments.

Preparing Revision Guidelines

For complete guidelines on revision requirements for your article type, please see the journal Article Types requirement at <https://journals.asm.org/journal/mSystems/article-types>. **Submissions of a paper that does not conform to mSystems guidelines will delay acceptance of your manuscript.**

Sincerely,

Laura Sanchez

Editor, mSystems

Journals Department
Reviewer comments:

Reviewer #1 (Comments for the Author):

This paper looks at the effect of deleting *E. coli* genes on the RNA and protein levels of the surrounding genes in an operon. The authors conclude that genes immediately downstream of a deletion often have reduced translation, and that this effect increases as the distance between the deletion and the downstream gene decreases. This is an interesting observation, and the work serves as a cautionary tale for experiments involving gene deletions. The paper relies heavily on large sets of proteomic and transcriptomic data. This approach has the advantage of considering genes in many different contexts, but has the disadvantage that the data are rather variable when comparing different deletions, and fairly noisy even when considering replicates for experiments using a single deletion strain. The authors argue that increasing the distance between a deletion and the downstream gene reduces the polar effect, but their own data are somewhat contradictory on this point. There is also no real mechanistic insight, and the authors do not offer a model to explain the observed polar effect.

Specific comments:

- Several of the gene deletions analyzed in Figure 1 are for genes encoding regulators that modulate expression of their own operons. These deletions likely add considerable variability to the dataset. I suggest removing data for genes encoding regulators known to modulate their own expression.
- Figure 2A makes the case that the distance between a deletion and the downstream gene impacts the magnitude of the polar effect. Figure 2C, right panel supports this conclusion, although the effect is modest, and the trend is heavily dependent on one gene (*hisG*). Figure 2C left panel does not support the conclusion. Given that Figures 2A and 2C left panel both involve marked

deletions, these data appear to be contradictory. Overall, I am not sufficiently convinced by these data that there is a distance-dependent effect on polarity.

- The data shown in Figure 2C are rather noisy, making them difficult to interpret. A reporter-based approach may be a better way to address the importance of the distance between a gene and the upstream deletion.

- The authors do not present a clear model for the polar effect they observe. They suggest that translational initiation or translational coupling may be involved, but they do not speculate how initiation or coupling could be affected by increasing the length of the upstream ORF.

- Figure 2. The nature of the data presented is not very clear. I assume these are proteomic data, but this is not indicated in the figure or legend.

- Line 145. "Although the protein levels of the downstream gene of Δ pabC, Δ hisG and Δ rfbB moved closer to wildtype levels when the leftover region was increased to 99 nucleotides". I do not agree with this statement for hisG or rfbB.

Reviewer #2 (Comments for the Author):

The single-gene deletion libraries represent a powerful tool used by many laboratories to characterize genes function in different bacteria and discover new essential targets for newly developed antibacterial drugs. Such libraries were designed to eliminate the function of specific individual genes and avoid any effects on the expression of other genes. In the presented work by Mateus et al., the authors show that the deletion of single genes located in operons can affect the expression level of the genes located downstream (but not upstream) of the deletion. This conclusion is based on the analysis of proteomic and RNA-seq data collected from 133 and 71 single deletion mutants of *Escherichia coli*, respectively. The results demonstrate that nearly 40% of deletions in operons affect (negatively or positively) the expression of the downstream genes by at least two times! Also, the authors show that the changes may arise from perturbations of the gene expression on transcriptional or translational levels. To my mind, the results presented by Mateus et al. are of great value since they urge researchers who use the single-gene deletion libraries to be careful in their conclusions and support them with some additional independent experiments.

Because such libraries are actively used by many laboratories worldwide, I believe that this study will be interesting for many journal readers.

I think that the manuscript represents a solid and comprehensive study and is acceptable for publication in *mSystems*. At the same time, some minor points should be addressed by the authors.

It is not clear from the text how many biological replicates of proteomic or RNA-seq experiments for each deletion mutant (and wildtype control) were included in the performed analysis. If there are no independent replicates, how the authors can address the possibility that their observation can result from some fluctuation in the expression of downstream genes. [For example, up and downregulation of 6 and 7 operons of 34 operons from proteomic analysis, respectively, as noticed in Line 94]. In this regard, it probably makes sense to move Figure S2 (where the authors did the same analysis as in Figure 1 for genes located in different operons as a deleted gene). To my mind, together with the absence of effect on the expression of upstream genes, this is important control showing that the gene deletion specifically affects the expression of other genes only when they are located in the same operon.

Figure S3, where the authors show a correlation in gene expression changes observed in deletion mutants on protein and RNA levels, is confused. The colors of dots, representing the position of the genes relative to deletion, on the plot are hard to distinguish. Do colors here make sense at all?

Probably there is a typo in Line 118: "Polar post-transcriptional effects on to the gene directly downstream of the deletion."

July 2021

mSystems (Impact factor 5-7)

“Transcriptional and post-translational polar effects in bacterial gene deletion libraries”

The single-gene deletion libraries represent a powerful tool used by many laboratories to characterize genes function in different bacteria and discover new essential targets for newly developed antibacterial drugs. Such libraries were designed to eliminate the function of specific individual genes and avoid any effects on the expression of other genes. In the presented work by Mateus et al., the authors show that the deletion of single genes located in operons can affect the expression level of the genes located downstream (but not upstream) of the deletion. This conclusion is based on the analysis of proteomic and RNA-seq data collected from 133 and 71 single deletion mutants of *Escherichia coli*, respectively. The results demonstrate that nearly 40% of deletions in operons affect (negatively or positively) the expression of the downstream genes by at least two times! Also, the authors show that the changes may arise from perturbations of the gene expression on transcriptional or translational levels. To my mind, the results presented by Mateus et al. are of great value since they urge researchers who use the single-gene deletion libraries to be careful in their conclusions and support them with some additional independent experiments.

Because such libraries are actively used by many laboratories worldwide, I believe that this study will be interesting for many journal readers.

I think that the manuscript represents a solid and comprehensive study and is acceptable for publication in mSystems. At the same time, some minor points should be addressed by the authors.

1. It is not clear from the text how many biological replicates of proteomic or RNA-seq experiments for each deletion mutant (and wildtype control) were included in the performed analysis. If there are no independent replicates, how the authors can address the possibility that their observation can result from some fluctuation in the expression of downstream genes. [For example, up and downregulation of 6 and 7 operons of 34 operons from proteomic analysis, respectively, as noticed in Line 94]. In this regard, it probably makes sense to move Figure S2 (where the authors did the same analysis as in Figure 1 for genes located in different operons as a deleted gene). To my mind, together with the absence of effect on the expression of upstream genes, this is important control showing that the gene deletion specifically affects the expression of other genes only when they are located in the same operon.
2. Figure S3, where the authors show a correlation in gene expression changes observed in deletion mutants on protein and RNA levels, is confused. The colors of dots, representing the position of the genes relative to deletion, on the plot are hard to distinguish. Do colors here make sense at all?
3. Probably there is a typo in Line 118: "Polar post-transcriptional effects on to the gene directly downstream of the deletion."

Best regards

We thank both reviewers for the insightful comments, which we have now addressed point-by-point below.

Reviewer #1 (Comments for the Author):

This paper looks at the effect of deleting *E. coli* genes on the RNA and protein levels of the surrounding genes in an operon. The authors conclude that genes immediately downstream of a deletion often have reduced translation, and that this effect increases as the distance between the deletion and the downstream gene decreases. This is an interesting observation, and the work serves as a cautionary tale for experiments involving gene deletions. The paper relies heavily on large sets of proteomic and transcriptomic data. This approach has the advantage of considering genes in many different contexts, but has the disadvantage that the data are rather variable when comparing different deletions, and fairly noisy even when considering replicates for experiments using a single deletion strain. The authors argue that increasing the distance between a deletion and the downstream gene reduces the polar effect, but their own data are somewhat contradictory on this point. There is also no real mechanistic insight, and the authors do not offer a model to explain the observed polar effect.

Specific comments:

1. Several of the gene deletions analyzed in Figure 1 are for genes encoding regulators that modulate expression of their own operons. These deletions likely add considerable variability to the dataset. I suggest removing data for genes encoding regulators known to modulate their own expression.

We checked all the knockouts included in Figure 1 (both from the proteomics and RNA-seq dataset) and identified 8 genes that encode transcriptional factors, which could potentially regulate their own operons ($\Delta cpxR$, Δfis , Δfur , $\Delta ihfA$, $\Delta ompR$, $\Delta phoP$, $\Delta rscB$, $\Delta rpoZ$). We have redrawn Figure 1b without these deletions and the overall effects are indistinguishable. We include this new figure as Figure S2 and have added a sentence in the results section: “These effects at the protein and transcript levels were evident even when excluding deletions of genes that may regulate the expression of their own operons (**Figure S2**).”

Figure S2. Operon polar effects are present even when excluding deletions of genes that may regulate the expression of their own operon. As in **Figure 1b**, excluding mutants of transcription factors which may regulate their own operons ($\Delta cpxR$, Δfis , Δfur , $\Delta ihfA$, $\Delta ompR$, $\Delta phoP$, $\Delta rscB$, $\Delta rpoZ$). Protein (76 deletion mutants): $n_{-4} = 5$, $n_{-3} = 9$, $n_{-2} = 18$, $n_{-1} = 46$, $n_1 = 57$, $n_2 = 32$, $n_3 = 19$, $n_4 = 10$. RNA (26 deletion mutants) (25): $n_{-2} = 7$, $n_{-1} = 18$, $n_1 = 18$, $n_2 = 10$, $n_3 = 6$, $n_4 = 5$.

Action taken: We now include a new supplementary figure and edited the text in the results section.

- Figure 2A makes the case that the distance between a deletion and the downstream gene impacts the magnitude of the polar effect. Figure 2C, right panel supports this conclusion, although the effect is modest, and the trend is heavily dependent on one gene (*hisG*). Figure 2C left panel does not support the conclusion. Given that Figures 2A and 2C left panel both involve marked deletions, these data appear to be contradictory. Overall, I am not sufficiently convinced by these data that there is a distance-dependent effect on polarity.

We politely disagree with the reviewer that there is no effect of the distance between two genes and the strength of “translational” downregulation of the gene downstream. Our strongest evidence comes from the large number of mutants covered and the strong correlation between these two parameters ($r_s=0.62$, $p=2.5 \times 10^{-4}$, $n=30$). However, we do agree that the mechanism for this effect is still unclear. Our results suggest that there is both a translation initiation component and a translational coupling component (and likely other mechanisms that we are unaware of) contributing to this effect. The individual contribution of each of these mechanisms to each case is likely variable and hence the analysis with the 5 mutants shown in Figure 2c has some variation. Although simply increasing the distance to downstream does not alleviate the problems for all mutants (Figure 2c, left panel), combining this with excising the resistance

cassette fully resolves the observed translational effect for 4 of the 5 mutants (except for $\Delta rfbB$). Further studies are necessary to fully elucidate the mechanism of this phenomenon (see the next answers).

Action taken: We have added the correlation between distance and downregulation to the results section.

3. The data shown in Figure 2C are rather noisy, making them difficult to interpret. A reporter-based approach may be a better way to address the importance of the distance between a gene and the upstream deletion.

The moderate noise in protein expression in the data shown in Figure 2c could be biological (protein expression being sensitive to growth stage) or could be due to secondary mutations that the two independently generated clones carry. That being said, the median variance is 1.03-fold (i.e., in most mutants the measurements are consistent between replicates), and the maximum variance is barely above 2-fold—both of which are acceptable to us. Transcriptional or translational reporters have also similar noise levels, and may introduce additional biases due to the non-native nature of fusion (add/eliminate elements of post-transcriptional regulation). Nevertheless, we have edited the text in the conclusion section to acknowledge these limitations: “This effect was correlated with the intergenic distance between the deleted gene and the downstream gene. Our follow-up work suggests that problems with translation initiation or lack of translational coupling lead to this effect, but further studies will be required to properly understand the underlying mechanism.”

Action taken: We have rewritten the conclusion to acknowledge that further work is needed.

4. The authors do not present a clear model for the polar effect they observe. They suggest that translational initiation or translational coupling may be involved, but they do not speculate how initiation or coupling could be affected by increasing the length of the upstream ORF.

We do not feel comfortable to speculate too much on this, and thus to provide a model with the data available. We can imagine multiple mechanisms being at play, for example:

- The ribosome might require sequences other than the canonical ribosome binding site to assemble and initiate translation (in *E. coli* only about 50% of the genes contain Shine-Dalgarno-like sequences, see e.g., PMID: 30085185). Therefore, the deletion of these sequences might impair the translation of the gene downstream.
- Translating a longer coding sequence can allow for a larger number of ribosomes to be translating the “scar ORF” (upon excision), and thus a higher chance for translational coupling (i.e., upon

reaching the stop codon, the ribosome starts translating the gene downstream) leading to higher levels of the gene downstream.

- Other mechanisms might be at play, such as co-translational folding; with a larger ORF providing a larger scaffold for the protein downstream to be properly folded and not degraded.

No action taken as we feel that further studies are required.

5. Figure 2. The nature of the data presented is not very clear. I assume these are proteomic data, but this is not indicated in the figure or legend.

We apologize for the omission. The data refer to proteomics data and this has been clarified in the figure and respective legend.

Action taken: We have changed figure 2 and its legend to clarify that this refers to proteomics data.

6. Line 145. "Although the protein levels of the downstream gene of $\Delta pabC$, $\Delta hisG$ and $\Delta rfbB$ moved closer to wildtype levels when the leftover region was increased to 99 nucleotides". I do not agree with this statement for hisG or rfbB.

The reviewer is correct, as we meant to write $\Delta pabC$, $\Delta surA$ and $\Delta oppB$. We apologize for the typo and have rewritten the sentence to: "Although the protein levels of the downstream gene of $\Delta pabC$, $\Delta surA$ and $\Delta oppB$ moved closer to wildtype levels when the leftover region was increased to 99 nucleotides".

Action taken: We have corrected this typo.

Reviewer #2 (Comments for the Author):

The single-gene deletion libraries represent a powerful tool used by many laboratories to characterize genes function in different bacteria and discover new essential targets for newly developed antibacterial drugs. Such libraries were designed to eliminate the function of specific individual genes and avoid any effects on the expression of other genes. In the presented work by Mateus et al., the authors show that the deletion of single genes located in operons can affect the expression level of the genes located downstream (but not upstream) of the deletion. This conclusion is based on the analysis of proteomic and RNA-seq data collected from 133 and 71 single deletion mutants of Escherichia coli, respectively. The results demonstrate that nearly 40% of deletions in operons affect (negatively or positively) the expression of the downstream genes by at least two times! Also, the authors show that the changes may

arise from perturbations of the gene expression on transcriptional or translational levels. To my mind, the results presented by Mateus et al. are of great value since they urge researchers who use the single-gene deletion libraries to be careful in their conclusions and support them with some additional independent experiments.

Because such libraries are actively used by many laboratories worldwide, I believe that this study will be interesting for many journal readers.

I think that the manuscript represents a solid and comprehensive study and is acceptable for publication in mSystems. At the same time, some minor points should be addressed by the authors.

1. It is not clear from the text how many biological replicates of proteomic or RNA-seq experiments for each deletion mutant (and wildtype control) were included in the performed analysis. If there are no independent replicates, how the authors can address the possibility that their observation can result from some fluctuation in the expression of downstream genes. [For example, up and downregulation of 6 and 7 operons of 34 operons from proteomic analysis, respectively, as noticed in Line 94]. In this regard, it probably makes sense to move Figure S2 (where the authors did the same analysis as in Figure 1 for genes located in different operons as a deleted gene). To my mind, together with the absence of effect on the expression of upstream genes, this is important control showing that the gene deletion specifically affects the expression of other genes only when they are located in the same operon.

We apologize for the lack of information on the number of replicates. We have used (when possible) two independent clones of the Keio collection for our analysis, with all mutants profiled at least in duplicate and their proteome changes averaged. To clarify, we did not use a WT control, but used the median expression of other mutants in the same mass spectrometry run as the control, as explained in more detail in Mateus et al., Nature (2020). We have now clarified these points in the results and methods, and made it more explicit in Table S1 which replicate the data originates from:

“When available, mutants were used directly from the Keio collection using two independent clones (with the exception of $\Delta atpD$, $\Delta atpE$, $\Delta clpP$, $\Delta dedD$, $\Delta rfaC$, for which the same clone was used in duplicate) (1, 17) (**Table S2**). [...] Then, for every protein, we calculated the ratio of the signal sum intensity of each mutant to the median signal sum of the same protein in all the mutants in the same mass spectrometry experiment (for data originating from Mateus et al. (11) we used the first two temperatures analyzed for each mutant). Fold-change data from replicates was averaged.”

We agree that Figure S2 is an important control that the effect is restricted to operons. However, we feel that it would be confusing to have it in Figure 1 (since this figure focuses on operon effects). Given the

size limitations of an “Observation” article (max. 2 figures) we think that this might be more appropriate as a supplementary figure. If the editor feels it is important to have it in main text, we are happy to move this figure to the main text as an extra figure.

Action taken: Clarified the number of replicates used in the results and methods sections and updated Table S1.

2. Figure S3, where the authors show a correlation in gene expression changes observed in deletion mutants on protein and RNA levels, is confused. The colors of dots, representing the position of the genes relative to deletion, on the plot are hard to distinguish. Do colors here make sense at all?

We agree with the reviewer that the colors were rather difficult to distinguish and they did not particularly add a meaningful message to the figure. We have thus removed them from the figure.

Action taken: Removed color coding from this figure (currently, Figure S4).

3. Probably there is a typo in Line 118: "Polar post-transcriptional effects on to the gene directly downstream of the deletion."

Action taken: We have now corrected the typo.

August 18, 2021

Dr. Mikhail Savitski
European Molecular Biology Laboratory
Heidelberg
Germany

Re: mSystems00813-21R1 (Transcriptional and post-transcriptional polar effects in bacterial gene deletion libraries)

Dear Dr. Mikhail Savitski:

Thank you for resubmitting the revised version of the manuscript. The two reviewers have re-reviewed the manuscript. I have included reviewer 1 comments should you wish to address the minor comment and to note one of the reviewers concerns (copied directly below). Given the nature of the observation, and more work is needed to tease out the mechanism of the finding which is stated in the manuscript, I believe the manuscript is in fine form for publication.

The authors have responded positively to my comments and those of the other reviewer, with the exception of my second comment. Specifically, I remain unconvinced by the current data that there is a distance-dependent effect on polarity. Figure 2A suggests there is a distance-dependent effect, but data in the left panel of Figure 2C are not consistent with this. Note that data in Figure 2A and the left panel of Figure 2C are directly comparable because the strains retain the antibiotic resistance cassette, so the disparity here is striking. Data in the right panel of Figure 2C (strains where the antibiotic resistance cassette has been removed) are more consistent with those in Figure 2A, but the effect is driven largely by a single strain (hisG deletion). As things stand, the data are intriguing, but insufficient to convince me of the effect.

I have one other minor comment (apologies for not raising this in my initial review): line 31 (abstract), I think speculating on the mechanism is not appropriate for the abstract given the lack of experimental data that speak to this question. Since the paper is an "Observation", I recommend simply stating the result here.

Your manuscript has been accepted, and I am forwarding it to the ASM Journals Department for publication. For your reference, ASM Journals' address is given below. Before it can be scheduled for publication, your manuscript will be checked by the mSystems senior production editor, Ellie Ghatineh, to make sure that all elements meet the technical requirements for publication. She will contact you if anything needs to be revised before copyediting and production can begin. Otherwise, you will be notified when your proofs are ready to be viewed.

As an open-access publication, mSystems receives no financial support from paid subscriptions and depends on authors' prompt payment of publication fees as soon as their articles are accepted. =

Publication Fees:

You will be contacted separately about payment when the proofs are issued; please follow the instructions in that e-mail. Arrangements for payment must be made before your article is published. For a complete list of **Publication Fees**, including supplemental material costs, please

visit our website.

We recognize that the video files can become quite large, and so to avoid quality loss ASM suggests sending the video file via <https://www.wetransfer.com/>. When you have a final version of the video and the still ready to share, please send it to Ellie Ghatineh at eghatineh@asmusa.org.

Sincerely,

Laura Sanchez
Editor, mSystems

Journals Department
Figure S4: Accept

Figure S5: Accept

Table S1: Accept

Figure S3: Accept

Figure S7: Accept

Figure S2: Accept

Figure S1: Accept

Source data: Accept

Table S2: Accept

Figure S6: Accept